# Short communication: Lifetime musical activity and resting-state functional connectivity in cognitive networks

**Maxie Liebscher**[1]*, Andrea Dell'Orco[1,2], Johanna Doll-Lee[3], Katharina Buerger[4,5], Peter Dechent[6], Michael Ewers[4], Klaus Fliessbach[7,8], Wenzel Glanz[9], Stefan Hetzer[10], Daniel Janowitz[5], Ingo Kilimann[11,12], Christoph Laske[13,14], Falk Lüsebrink[9], Matthias Munk[13,15], Robert Perneczky[4,16,17,18], Oliver Peters[19,20], Lukas Preis[20], Josef Priller[19,21,22,23], Boris Rauchmann[16,24,25], Ayda Rostamzadeh[26], Nina Roy-Kluth[7], Klaus Scheffler[27], Anja Schneider[7,8], Björn H. Schott[28,29], Annika Spottke[7,30], Eike Spruth[19,21], Stefan Teipel[11,12], Jens Wiltfang[28,29,31], Frank Jessen[26,32,33], Emrah Düzel[9,34], Michael Wagner[7,8], Sandra Röske[7], Miranka Wirth[1‡]*, On behalf of DELCODE study group[¶]

**1** German Center for Neurodegenerative Diseases (DZNE), Dresden, Germany, **2** Department of Neuroradiology, Charité –Universitätsmedizin Berlin, Corporate Member of Freie Universität Berlin and Humboldt- Universität zu Berlin, Berlin, Germany, **3** Department of Neurology, Hannover Medical School, Hannover, Germany, **4** German Center for Neurodegenerative Diseases (DZNE), Munich, Germany, **5** Institute for Stroke and Dementia Research (ISD), University Hospital, LMU Munich, Munich, Germany, **6** Department of Cognitive Neurology, MR-Research in Neurosciences, Georg-August-University Goettingen, Göttingen, Germany, **7** German Center for Neurodegenerative Diseases (DZNE), Bonn, Germany, **8** Department of Neurodegenerative Disease and Geriatric Psychiatry/Psychiatry, University of Bonn Medical Center, Bonn, Germany, **9** German Center for Neurodegenerative Diseases (DZNE), Magdeburg, Germany, **10** Center for Advanced Neuroimaging, Charité –Universitätsmedizin Berlin, Berlin, Germany, **11** German Center for Neurodegenerative Diseases (DZNE), Rostock, Germany, **12** Department of Psychosomatic Medicine, Rostock University Medical Center, Rostock, Germany, **13** German Center for Neurodegenerative Diseases (DZNE), Tübingen, Germany, **14** Section for Dementia Research, Hertie Institute for Clinical Brain Research and Department of Psychiatry and Psychotherapy, University of Tübingen, Tübingen, Germany, **15** Department of Psychiatry and Psychotherapy, University of Tübingen, Tübingen, Germany, **16** Department of Psychiatry and Psychotherapy, University Hospital, LMU Munich, Munich, Germany, **17** Munich Cluster for Systems Neurology (SyNergy) Munich, Munich, Germany, **18** Ageing Epidemiology Research Unit (AGE), School of Public Health, Imperial College London, London, United Kingdom, **19** German Center for Neurodegenerative Diseases (DZNE), Berlin, Germany, **20** Institute of Psychiatry and Psychotherapy, Charité –Universitätsmedizin Berlin, Corporate Member of Freie Universität Berlin and Humboldt-Universität zu Berlin, Berlin, Germany, **21** Department of Psychiatry and Psychotherapy, Charité –Universitätsmedizin Berlin, Berlin, Germany, **22** Department of Psychiatry and Psychotherapy, School of Medicine, Technical University of Munich, Munich, Germany, **23** University of Edinburgh and UK DRI, Edinburgh, United Kingdom, **24** Sheffield Institute for Translational Neuroscience (SITraN), University of Sheffield, Sheffield, United Kingdom, **25** Department of Neuroradiology, University Hospital LMU, Munich, Germany, **26** Department of Psychiatry, University of Cologne, Medical Faculty, Cologne, Germany, **27** Department for Biomedical Magnetic Resonance, University of Tübingen, Tübingen, Germany, **28** Department of Psychiatry and Psychotherapy, University Medical Center Goettingen, University of Goettingen, Goettingen, Germany, **29** German Center for Neurodegenerative Diseases (DZNE), Goettingen, Germany, **30** Department of Neurology, University of Bonn, Bonn, Germany, **31** Department of Medical Sciences, Neurosciences and Signaling Group, Institute of Biomedicine (iBiMED), University of Aveiro, Aveiro, Portugal, **32** Excellence Cluster on Cellular Stress Responses in Aging-Associated Diseases (CECAD), University of Cologne, Cologne, Germany, **33** German Center for Neurodegenerative Diseases (DZNE), Cologne, Germany, **34** Institute of Cognitive Neurology and Dementia Research (IKND), Otto-von-Guericke University, Magdeburg, Germany

‡ MW share last authorship on this work.
¶ The members of the DELCODE study group are listed in the acknowledgments.
* miranka.wirth@dzne.de (MW); maxie.liebscher@dzne.de (ML)

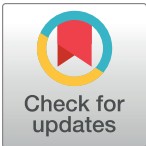

**Data Availability Statement:** Due to the sensitive and personal nature of the data, we are unable to publicly disclose the data. The anonymized data used for this study will be made available by

request from any qualified investigator through the DZNE-DELCODE Steering Board for purposes of replicating procedures and results. Requests to access the minimal dataset should be directed to the German Center for Neurodegenerative Diseases (DZNE), Bonn. For contact information please refer to: https://www.dzne.de/en/research/studies/clinical-studies/delcode/ (Study Coordination and Project Management). We used existing data analysis packages for the neuroimaging and statistical analysis of this study. All relevant R scripts used for data analysis are freely available via the open science framework (OSF, https://osf.io/g73a9/).

**Funding:** The overall DELCODE study was funded by the German Center for Neurodegenerative Diseases (Deutsches Zentrum für Neurodegenerative Erkrankungen [DZNE]), reference number: BN012. For the present study, the authors received no specific funding. The funders had no role in study design, data collection and analysis, decision to publish, or preparation of the manuscript.

**Competing interests:** O. Peters received fees for consultation from Abbvie, Biogen, Eisai, Griffols, MSD Roche, and Schwabe. J. Priller received fees for consultation, lectures, and patents from Neurimmune, Axon, Desitin, and Epomedics. J. Wiltfang is an advisory board member of Abbott, Biogen, Boehringer Ingelheim, Immunogenetics, Lilly, MSD Sharp & Dohme, and Roche Pharma and received honoraria for lectures from Actelion, Amgen, Beeijing Yibai Science and Technology Ltd., Janssen Cilag, Med Update GmbH, Pfizer, Roche Pharma and holds the following patents: PCT/EP 2011 001724 and PCT/EP 2015 052945. J. Wiltfang is supported by an Ilidio Pinho professorship, iBiMED (UIDB/04501/2020) at the University of Aveiro, Portugal. E. Düzel received fees for consultation from Roche, Biogen, RoxHealth and holds shares in neotiv. F. Jessen received fees for consultation from Eli Lilly, Novartis, Roche, BioGene, MSD, Piramal, Janssen, and Lundbeck. The remaining authors report no conflicts of interest. This does not alter our adherence to PLOS ONE policies on sharing data and materials.

# Abstract

## Background

Participation in multimodal leisure activities, such as playing a musical instrument, may be protective against brain aging and dementia in older adults (OA). Potential neuroprotective correlates underlying musical activity remain unclear.

## Objective

This cross-sectional study investigated the association between lifetime musical activity and resting-state functional connectivity (RSFC) in three higher-order brain networks: the Default Mode, Fronto-Parietal, and Salience networks.

## Methods

We assessed 130 cognitively unimpaired participants ($\geq$ 60 years) from the baseline cohort of the DZNE-Longitudinal Cognitive Impairment and Dementia Study (DELCODE) study. Lifetime musical activity was operationalized by the self-reported participation in musical instrument playing across early, middle, and late life stages using the Lifetime of Experiences Questionnaire (LEQ). Participants who reported musical activity during all life stages ($n = 65$) were compared to controls who were matched on demographic and reserve characteristics (including education, intelligence, socioeconomic status, self-reported physical activity, age, and sex) and never played a musical instrument ($n = 65$) in local (seed-to-voxel) and global (within-network and between-network) RSFC patterns using pre-specified network seeds.

## Results

Older participants with lifetime musical activity showed significantly higher local RSFC between the medial prefrontal cortex (Default Mode Network seed) and temporal as well as frontal regions, namely the right temporal pole and the right precentral gyrus extending into the superior frontal gyrus, compared to matched controls. There were no significant group differences in global RSFC within or between the three networks.

## Conclusion

We show that playing a musical instrument during life relates to higher RSFC of the medial prefrontal cortex with distant brain regions involved in higher-order cognitive and motor processes. Preserved or enhanced functional connectivity could potentially contribute to better brain health and resilience in OA with a history in musical activity.

## Trial registration

German Clinical Trials Register (DRKS00007966, 04/05/2015).

# 1 Introduction

## 1.1 Musical activity and late-life cognition

Long-term participation in playing a musical instrument, as an integrated multimodal lifestyle activity, could be protective against cognitive decline and dementia in late life. A history of musical activity in older adults (OA) is associated with reduced risk of developing dementia [1]. Cross-sectional studies have further shown that musically active OA (both amateurs and professionals) have better cognitive function across multiple domains compared to control participants [2–6]. Likewise, training/intervention studies in musically non-experienced OA have demonstrated positive effects of musical instrument playing on cognitive abilities compared to control interventions [7–11]. Playing a musical instrument may therefore help improve or maintain brain and cognitive health in late life [12, 13], however, possible neuroprotective correlates underlying musical activity in OA need further investigation.

## 1.2 Musical activity and functional brain connectivity

It has been suggested that participation in musical activities promotes neural plasticity in motor, sensory, and cognitive networks throughout life [14–16]. Playing a musical instrument necessitates the activation and integration of multimodal motor, sensory, cognitive and emotional processes, which may help to preserve neural function in distributed brain networks into older age. Studies in cognitively unimpaired OA have shown that greater musical activity/experience is associated with greater neural resources or neural capacities in distant brain regions including higher-order frontal, temporal, and/or parietal areas [6, 17]. This aligns with findings in young to middle-aged musicians, showing greater functional and/or structural connectivity within (but not limited to) frontal-temporal-parietal brain networks compared to control participants [18–20]. In a recent study specifically addressing OA, greater musical experience was reported to correlate with variations in resting-state functional connectivity (RSFC) of insular regions with sensory, motor and cognitive brain regions, including the precentral and postcentral gyrus as well as the prefrontal cortex [21].

Despite these findings, the association between musical activity and RSFC in large-scale brain systems that are vulnerable to both normal and pathological neurocognitive aging processes remains unknown. This association is important to investigate, because preserved neural activation and connectivity in higher-order brain networks is considered to act as a neural correlate of cognitive reserve and resilience in older age [22–25]. Previous research points to the particular importance of three large-scale resting-state networks (RSN, also known as triple networks) that have been shown to be involved in neurocognitive aging and Alzheimer's disease (AD) [26], namely the Fronto-Parietal Network (FPN, also known as central executive network), the Default Mode Network (DMN), and the Salience Network (SAL) [27–29]. In these higher-order brain networks, lower RSFC has been shown to be associated with lifestyle risk factors and/or decline of cognitive abilities in OA [26, 30]. In contrast, higher RSFC in these brain systems (namely the FPN and SAL) has been related to protective lifestyle factors and/or preserved cognitive abilities in the context of brain pathology [22, 31], supporting a contribution of these neural correlates in brain reserve and resilience.

## 1.3 The present study

In this cross-sectional study, we aimed to investigate to what extent self-reported playing of a musical instrument in early, middle, and late life stages (subsequently termed "lifetime musical activity") might be related to RSFC within the FPN, DMN, SAL networks in cognitively unimpaired OA. As mentioned above, previous studies have shown that neural correlates associated

with musical activity/experience in OA partially overlap with these higher-order brain networks [6, 17, 21]. We therefore addressed the hypothesis that participation in this multimodal lifestyle activity relates to increased RSFC in the triple networks. More specifically, we compared local (seed-to-voxel) and global (within- and between-network) RSFC patterns in participants ($\geq$ 60 years) from the DZNE–Longitudinal Cognitive Impairment and Dementia Study (DELCODE) cohort [32] who reported having played a musical instrument during all life stages compared to matched control participants who never played a musical instrument [6].

## 2 Material and methods

This study was based on the baseline dataset of the DZNE–DELCODE cohort, an ongoing German multicenter longitudinal study [32] designed and carried out in accordance with the ethical principles of the Declaration of Helsinki. The protocol was approved by local ethical committees at each study site and all participants provided written informed consent. The DELCODE study is registered at the German Clinical Trials Register (DRKS00007966; date: 04/05/2015). The authors did not have access to detailed information to identify individual participants. Detailed information on the study protocol is provided elsewhere [32].

### 2.1 Participants

Based on our hypothesis, we selected cognitively unimpaired participants from the DZNE--DELCODE baseline cohort [32], including healthy control participants (HC), participants with subjective cognitive decline (SCD), and participants with a family history (FH) of Alzheimer's disease (AD). To assess cognitive functioning in these participants, the Consortium to Establish a Registry for Alzheimer's Disease (CERAD) neuropsychological test battery [33] was applied at all assessment sites. Normal or unimpaired cognitive performance was defined by a test performance within –1.5 standard deviations (*SD*) of the age, sex, and education-adjusted norms on all subtests of the CERAD neuropsychological battery. In addition to standard neuropsychological tests, neurological examinations and blood tests were carried out to exclude other diseases and conditions that could affect the participants' cognition. Further information are provided in the **S1 File** and the DELCODE study protocol [32]. All participants of the DELCODE cohort have fluent German skills and were 60 years or older.

The total sample of the present study included 130 participants, consisting of a group with lifetime musical activity (*n* = 65) and a group of matched controls (*n* = 65; **Table 1**). The latter group was selected from the DELCODE cohort using a one-to-one matching procedure based on pre-defined characteristics including socioeconomic status (SES), crystallized intelligence, self-reported physical activity, sex, age, years of education, and diagnostic group. The selection flow chart (**S1 Fig in S1 File**) and details on the selection process and one-to-one matching procedure can be found in the **S1 File**. The present sample and selection procedures overlapped with that of our previous study [6].

### 2.2 Measurements

**2.2.1 Self-reported assessment of musical activity.** Information about lifetime experiences of musical activity were derived from the Lifetime of Experiences Questionnaire (LEQ) [34] using a version adapted to the German population (LEQ-D) [35]. We used the same categorization scheme as described in detail in our previous study [6]. Briefly, participants answered questions about playing a musical instrument during their lifetime ("How often did you play a musical instrument?", 6-point Likert-scale: 0: never, 1: less than 1 time per month, 2: 1 time per month, 3: 2 times per month, 4: weekly, 5: daily) across the life stages (13–30 years, 30–65 years, and if applicable $\geq$ 65 years). Participants who reported that they had

**Table 1. Descriptive data of the matched sample (n = 130).**

| | Musical activity (*n* = 65) *M (SD)* | Matched controls (*n* = 65) *M (SD)* | Test statistics | *p* |
|---|---|---|---|---|
| Sex: female/male (*n*) | 28/37 | 30/35 | $\chi^2$ = 0.031 | .860 |
| Age (years) | 68.12 (6.78) | 68.45 (5.36) | *t = 0.301* | .764 |
| Education (years) | 16.45 (2.54) | 16.46 (2.65) | *t = 0.034* | .973 |
| Crystallized Intelligence [a] | 33.38 (2.10) | 33.43 (1.79) | *t = 0.135* | .893 |
| SES [b] | 67.84 (14.94) | 65.66 (18.65) | t = -0.735 | .464 |
| Diagnostic group: HC/SCD/FH (*n*) | 16/42/7 | 17/39/9 | $\chi^2$ = 0.391 | .822 |
| Lifetime physical activity [c] | 4.21 (0.76) | 4.25 (0.86) | *t = 0.270* | .788 |
| Current physical activity [d] | 34.34 (11.67) | 32.30 (11.53) | *t = -0.985* | .327 |
| LEQ total specific score [e] | 83.36 (13.61) | 79.24 (16.64) | *t = -1.359* | .177 |
| LEQ total non-specific score (adapted) [f] | 20.25 (3.86) | 18.54 (3.61) | *t = -1.709* | .090 |

*** p < .001

** p < .005

* p < .05.

M = mean, SD = standard deviation.

[a] Measured with the Multiple-Choice Vocabulary Test (MWT-B, minimum: 0, maximum: 37). Higher scores indicate higher intelligence.

[b] Measured by the International Socioeconomic Index of Occupational Information (ISEI; minimum: 16, maximum: 90), based on occupational information provided by participants. Higher scores indicate higher SES.

[c] Measured using the Lifetime of Experiences Questionnaire (LEQ), in which participants answered questions about the frequency of physical activity on a six-point Likert scale (0: never to 5: daily). Higher scores indicate more frequent physical activity.

[d] Measured using the Physical Activity Scale for the Elderly (PASE; minimum: 0). Higher scores indicate greater levels of current physical activity.

[e] Measured using the specific score of the LEQ that includes information on educational attainment and occupational complexity between the ages of 13 to > 65. Higher scores indicate higher cognitive enrichment.

[f] Measured using the non-specific score of the LEQ that includes information on leisure time activities between the ages of 13 to > 65. Higher scores indicate greater engagement in leisure time activities. Note, the score was adapted to exclude musical activity.

Key. AD: Alzheimer's disease, FH: AD family history; HC: healthy controls, SCD: subjective cognitive decline; SES: socioeconomic status.

played a musical instrument at all life stages, taking into account their respective age, and with a high frequency ($\geq$ 2 times per month) in at least one of these life stages, were categorized as having a lifetime history of musical activity. In contrast, participants who reported to never had played a musical instrument at any given life stage were classified as control group, which was matched in terms of demographic characteristics and reserve proxies (see above). Detailed information on the frequency of playing a musical instrument across the different life stages of each participant in the lifetime musical activity group is provided in our **S2 Fig in S1 File.**

**2.2.2 MRI acquisition, pre-processing and denoising.** The magnetic resonance imaging (MRI) data were acquired with nine 3.0 Tesla Siemens scanners (one Prisma system, one Skyra system, three TIM Trio systems, four Verio systems) at nine DZNE sites using harmonized acquisition protocols [32]. For the present study, we used structural T1-weighted (MPRAGE) and resting-state functional MRI (fMRI) data. Detailed sequence parameters can be found in the **S1 File**.

The MRI images were preprocessed using the default preprocessing pipeline for volume-based analyses (direct normalization to Montreal Neurological Institute (MNI)-space) of the CONN Functional Connectivity Toolbox version 18b [36] in the graphical user interface, based on SPM12 [37] and implemented in MATLAB [38]. In brief, this preprocessing pipeline includes functional realignment and unwarping, slice-timing correction, outlier identification,

and direct segmentation and normalization of the functional and structural images. The pipeline performs a non-linear transformation of the structural and functional MRI data to MNI space using the unified segmentation approach [39] that is based on the separation of the images in gray matter, white matter, and cerebrospinal fluid (CSF). As a result, the functional and structural images are located in the same space without being explicitly co-registered to each other. We performed a visual quality assurance in CONN to assess the quality of co-registration by checking that the anatomical and functional images are aligned with the MNI template. Lastly, the functional images were smoothed with an 8-mm full-width half-maximum Gaussian kernel. Detailed information on the default preprocessing pipeline is available in the CONN manual [40] and the online documentation [41].

We further applied the default denoising pipeline implemented in CONN (version 18b), which combines two steps to remove noise, outliers, and motion artifacts that can cause distortions in the functional MRI data. The first step in this denoising pipeline is a linear regression of potential confounding effects in the Blood Oxygenation Level Dependent (BOLD) signal using Ordinary Least Squares. This is followed by the removal of slow-frequency fluctuations with a band pass-filter of [0.008–0.09 Hz]. Detailed information on the default denoising pipeline is available in the CONN manual [40] and the online documentation [42].

**2.2.3 Functional MRI processing.** To assess local RSFC as well as global RSFC for each of the RSN (DMN, SAL, FPN), we selected network-specific region-of-interests (ROIs) provided by the CONN toolbox. These ROIs were obtained from an independent component analysis on 497 participants of the Human Connectome Project [36, 43]. For future replication studies, the network-specific ROIs used for the RSFC analysis and corresponding MNI centroid coordinates are provided in the **S1 File**. Next, we created custom seeds with an 8-mm sphere for each RSN using the network-specific ROIs with their respective MNI centroid coordinates, as provided by CONN.

In the context of our study, we assessed "local RSFC" as the functional coupling between a central node (i.e. seed region) of each RSN and other brain regions. The local RSFC was computed for each RSN by the a-priori selection of one ROI (or network seed) for each RSN that was of particular interest for our study. More precisely, we chose frontal seeds for each RSN, namely, for the DMN: medial prefrontal cortex seed (MPFC; +1, +55, -3), SAL: anterior cingulate cortex seed (ACC; 0, +22, +35), and FPN: left lateral prefrontal cortex seed (LPFC; -43, +33, +28). A visualization of seed locations is provided in S3 Fig in S1 File. The selection of frontal seeds for the RSN was based on previous research, showing an involvement of higher-order frontal regions in musical activity/training [6, 20, 21], cognitive reserve processes [22, 44] and cognitive functions associated with aging and AD [45–48]. Local RSFC values were calculated using the seed-to-voxel connectivity analysis performed in CONN, similar to our previous studies [31, 49]. First-level whole-brain seed-based connectivity maps were generated for each RSN and each participant by calculating Fisher's r-to-z-transformed correlation coefficients between the mean BOLD time course of each custom seed and the BOLD time course of all other voxels in the whole brain. The individual seed-based connectivity maps for each RSN were then subjected to a second-level analysis (see below).

We further assessed "global RSFC" defined as the functional coupling across spatially distant nodes within each RSN. To quantify global RSFC within each RSN (i.e., intra-network RSFC), we used the network-specific ROIs provided by CONN (see **S1 Table in S1 File**). The ROI-to-ROI connectivity values were computed for each pair of ROIs belonging to the respective network based on the pairwise BOLD signal correlations, which were then converted to *z-scores* using Fisher's r-to-z transformation. Using the automated script "*conn_withinbetweenROItest*" implemented in CONN, the average functional connectivity for each RSN and each participant

was computed. This measure of intra-network RSFC was extracted from CONN and subjected to statistical analysis.

In a post-hoc analysis, we additionally evaluated the global RSFC between the RSN (i.e., inter-network RSFC). We used the above-mentioned network-specific ROIs and the automated script "*conn_withinbetweenROItest*" as implemented in CONN to compute the inter-network RSFC for each possible network combination (DMN-SAL, DMN-FPN, SAL-FPN). The inter-network RSFC was calculated by averaging the ROI-to-ROI connectivity values (*z-scores*) over all given ROIs of each RSN combination for each participant. The resulting measure of inter-network RSFC was subsequently extracted from CONN for statistical analysis.

**2.2.4 Additional measures.**   The SES, crystallized intelligence, self-reported physical activity, sex, age, years of education, and diagnostic group were considered in the one-to-one matching procedure to select a matched control group (i.e., without history of musical instrument playing). Most of these measures were provided by the DELCODE baseline dataset [32].

As described in detail in our previous study [6], the SES was assessed using the International Socioeconomic Index of Occupational Information (ISEI, min. score: 16, max. score: 90) [50], with higher scores corresponding to higher SES. Briefly, the self-reported occupational history of each participant, as assessed by the LEQ across 10 five-year intervals from middle-to-late adulthood (age 30 to 79 years), was coded into occupational categories using the O*Net code system [51, 52]. The O*Net scores were then converted into ISEI scores using fully-automated publicly-available crosswalk procedures that included conversion to Standard Occupational Classification codes (SOC), International Standard Classification of Occupations (ISCO-08) and ISEI calculation [53]. The resulting ISEI scores were averaged across the given time intervals to calculate a mean SES measure for each participant.

Crystallized intelligence was estimated for each participant using the Multiple-Choice Vocabulary Intelligence Tests (MWT) [54]. Lifetime physical activity was assessed using participants' responses in the LEQ [34]. In addition, we assessed current physical activity (in the past 7 days) using the self-reported Physical Activity Scale for the Elderly (PASE) [55]. Further information is provided in the **S1 File**. Finally, we evaluated the total specific and non-specific (excluding musical activity) LEQ scores [34], which includes information about educational attainment and occupational complexity as well as participation in leisure time activities between the ages of 13 to > 65 (if applicable), respectively.

## 2.3 Statistical analysis

Statistical analyses were carried out in CONN or in R (version 4.1.2) [56] using RStudio (version 2022.02.0) [57]. The statistical models carried out in CONN are described below. For those analyses carried out in R, *p*-values of < .005 were considered statistically significant in agreement with Benjamin and colleagues [58]. All statistical assumptions were checked visually or by statistical tests (e.g., Shapiro-Wilk test, Breusch-Pagan test) before analyses were performed. Lastly, all plots were generated using the R package *ggplot2* (version 3.3.5) [59] and *ggalluvial* (version 0.12.5) [60] for additional plots in the **S1 File**.

**2.3.1 Sample characteristics.**   Participants with lifetime musical activity were compared with matched controls for selected demographic and behavioral data using the R package *psych* (version 2.2.3) [61], Student's t-test, and Pearson's Chi-squared test ($\chi^2$). Whenever the normal distribution requirement of the t-test was not met, Mann-Whitney-U-tests or Wilcoxon-tests (*W*) were performed.

**2.3.2 Local functional connectivity.**   To assess the association between lifetime musical activity (group variable) and local RSFC of each RSN seed, we performed a second-level whole-brain linear regression analysis with lifetime musical activity as independent variable,

the individual Fisher-z-transformed seed-connectivity maps as dependent variable and scanner location (dummy coded with reference) as co-variate of no interest. The statistical parametric map obtained for each RSN seed was thresholded using a voxel-level threshold of $p < .005$ (uncorrected), and a cluster extend threshold of $p < .05$ corrected for multiple comparisons using False Discovery Rate (FDR). The threshold procedure is comparable to our previous studies [31, 49] and is considered accetable for the detection of smaller effects [62]. The FSL image viewer (version 6.0.4) [63] was used to determine the center of gravity coordinates of the resulting clusters, which were then used in conjunction with the Harvard-Oxford Cortical Structural Atlas (RRID:SCR_001476) [64, 65] to determine the cluster locations. Mean local RSFC values (z-scores) were extracted for each significant cluster and each participant for follow-up analysis and visualization carried out in R (version 4.1.2) [56].

**2.3.3 Global functional connectivity.**   To assess the association between lifetime musical activity (group variable) and global RSFC within each RSN as well as the inter-network RSFC, we used the extracted global RSFC values for each RSN and each participant and performed multiple linear regression analysis in R (version 4.1.2) [56] with lifetime musical activity as the independent variable, the individual global RSFC values (z-scores) or inter-network RSFC values (z-scores) as dependent variable, and scanner location (see above) as co-variate of no interest.

# 3 Results

## 3.1 Sample characteristics

The final sample consisted of 130 participants with 65 participants in each group (lifetime musical activity: $n = 65$, matched controls: $n = 65$) with no significant differences in the group characteristics including demographic and behavioral measures. Descriptive data can be found in **Table 1**. In a post-hoc assessment, we further compared the two groups across additional characteristics and found no significant difference in social activities ($p = .783$) measured using the LEQ [34], personality traits of openness ($p = .957$), extraversion ($p = .926$), agreeableness ($p = .856$), neuroticism ($p = .909$), and conscientiousness ($p = .606$) measured using the 10-item short form of the Big Five Inventory (BFI-10) [66], subclinical depression ($p = .127$) measured using the 15-item short form of the Geriatric Depression Scale (GDS) [67], and the body mass index (BMI, $p = .305$) as an indicator of dietary habits.

## 3.2 Local functional connectivity

Results of the local RSFC analysis are provided in **Table 2** and **Fig 1**. In the DMN, participants with lifetime musical activity exhibited greater local functional connectivity between the MPFC (DMN seed) and two clusters in the right temporal pole (standardized β = 0.471) and the right precentral gyrus extending into the right superior frontal gyrus (β = 0.360). For the FPN and SAL seeds, there were no significant group differences in the seed-to-voxel connectivity patterns between participants with versus without lifetime musical activity.

## 3.3 Global functional connectivity

Results of the global (within-network) RSFC analysis are provided in the **(S3 Fig in S1 File)**. There were no significant group differences between participants with lifetime musical activity and matched controls in global (within-network) connectivity in the DMN (β = 0.113, $p = .220$), SAL (β = -0.006, $p = .946$) or FPN (β = -0.086, $p = .365$).

In a post-hoc evaluation, we further compared the inter-network RSFC between two groups. We found no significant differences between participants with lifetime musical activity

**Table 2. Resulting clusters of the seed-to-voxel connectivity analysis for the DMN seed.**

| No. Cluster | Center of gravity | Cluster | | Label | No. BA | Hemisphere |
|---|---|---|---|---|---|---|
| | MNI coordinates (x, y, z) | p-value (FDR-corrected) | Size (No. voxels) | (Percentage of overlap) | | |
| 1 | +33, +11, -26 | < .011* | 549 | Temporal pole (88%) | BA 38 (31%), BA 47 (28%) | right |
| 2 | +26, -11, +63 | < .033* | 391 | Precentral gyrus (35%), Superior frontal gyrus (21%) | BA 6 (51%), BA 4 (22%) | right |

*** p < .001

** p < .01

* p < .05.

Musical activity was used as predictor and dummy coded with 1 for lifetime musical activity and 0 for controls (no lifetime musical activity).

The statistical model was adjusted for scanner location (dummy coded) and results were corrected for multiple comparisons using FDR.

Corresponding brain regions were defined with the FSL image viewer using the center of gravity coordinates of the resulting clusters and the Harvard-Oxford Cortical Structural Atlas.

BA: Brodmann area, DMN: Default Mode Network, FDR: False Discovery Rate, MNI: Montreal Neurological Institute.

and controls in the inter-network RSFC of DMN-SAL ($\beta$ = -0.068, $p$ = .473), DMN-FPN ($\beta$ = 0.017, $p$ = .854), and SAL-FPN ($\beta$ = -0.025, $p$ = .786).

# 4 Discussion

## 4.1 Summary of findings

This cross-sectional study investigated the association between self-reported participation in lifetime musical activity and RSFC in the three higher-order brain networks that are vulnerable to aging and dementia, namely the DMN, SAL, and FPN (also known as triple networks). We compared cognitively unimpaired groups of participants with musical activity during all life stages and matched controls from the DZNE-DELCODE cohort [32] in terms of their local and global functional connectivity patterns. We show that playing a musical instrument during life is associated with higher RSFC of the MPFC with distant frontal and temporal regions that are partially associated with the DMN. Similar results were not obtained for local RSFC patterns of the SAL and FPN or for the global (within-network and between-network) RSFC patterns of the triple networks. Preserved or enhanced functional connectivity between distant higher-order brain regions could potentially contribute to better brain health and resilience in OA with a history of musical activity–a hypothesis that should be investigated in future studies.

## 4.2 Musical activity and local functional connectivity

Our study demonstrates that lifetime musical activity relates to higher functional connectivity between brain regions that are associated with higher-order cognitive and motor processes. Specifically, we show that older participants who reported playing a musical instrument during life have higher local RSFC between the MPFC (DMN seed) and two clusters in distant temporal and frontal regions (namely, the right temporal pole, and the right precentral extending into the superior frontal gyrus) compared to matched controls. In general, this result is consistent with previous cross-sectional studies showing that greater musical activity or musical experience in younger and older adults (amateurs and professionals) is related to mostly enhanced RSFC comprising large-scale brain networks, including sensory, motor, and cognitive regions [20, 21, 68]. Interestingly, a preliminary study [68] has demonstrated an increase

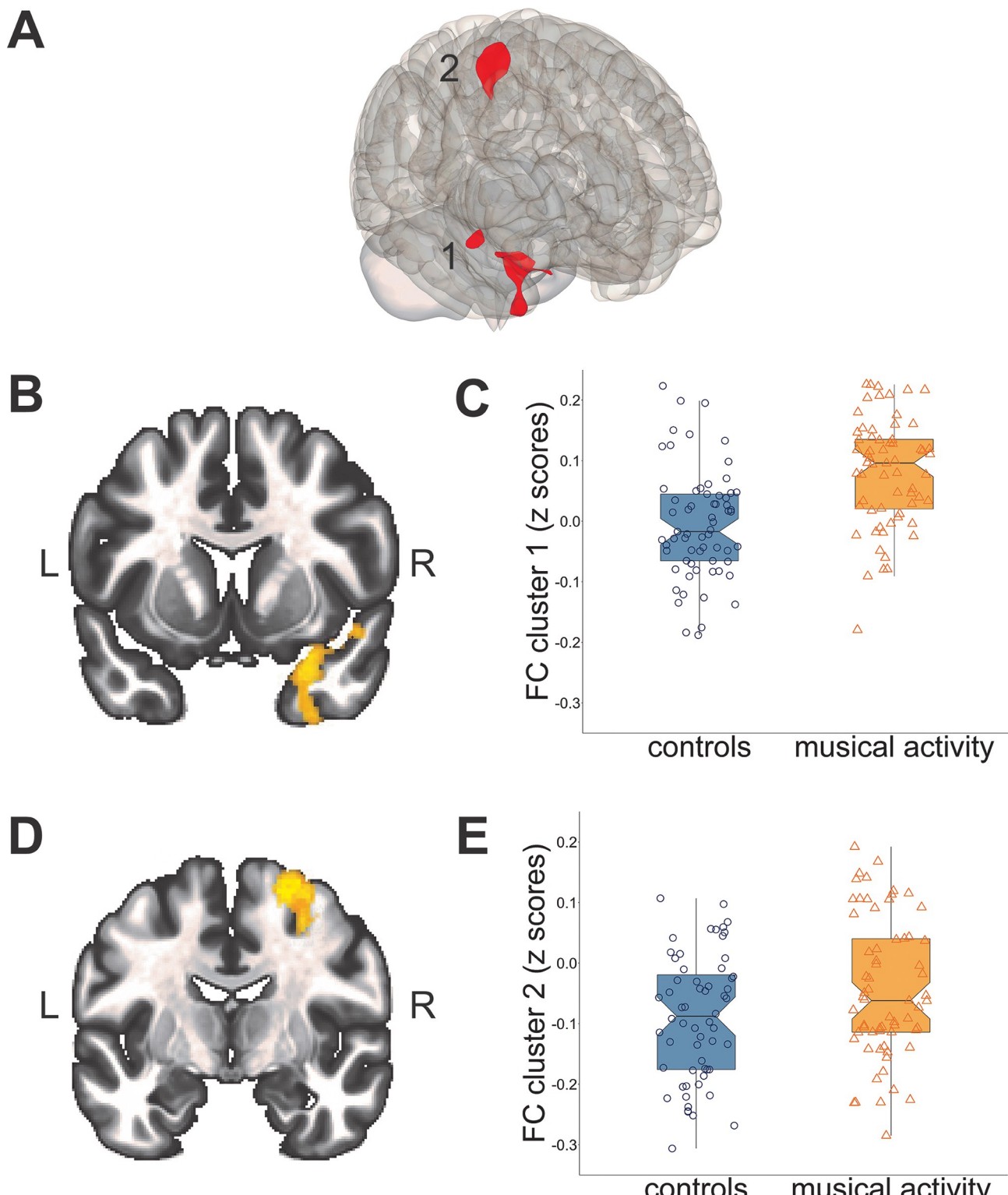

**Fig 1. Results of the local (seed-to-voxel) connectivity analysis. A.** The brain map shows the two significant clusters (1 and 2, displayed in red) resulting from seed-to-voxel analysis between the MPFC (DMN seed, displayed in the S3 Fig in S1 File) and all other voxels. The brain maps **(left side)** show the clusters (voxel-level threshold of p < 0.005 and cluster-level threshold of p < 0.05, FDR-corrected) located in temporal and frontal brain regions (L: left and R: right). **B.** Cluster 1: located in the right temporal pole (coronal plane, y = 8). **D.** Cluster 2: located in the right precentral gyrus extending to the superior frontal gyrus (coronal plane, y = -8). **C. and E.** The corresponding graphs **(right side)** show higher functional connectivity

(mean z scores) between the MPFC (DMN seed) and each cluster in older participants with lifetime musical activity compared to matched controls. Box plots display the median with 95% confidence intervals, interquartile range with lower (25th) and upper percentiles (75th), and individual data points within each group. **Key**: DMN: Default Mode Network, FC: Functional connectivity, FDR: False Discovery Rate, MPFC: Medial Prefrontal Cortex.

in mean RSFC with higher age in a small sample of musicians (*n* = 8, aged 20 to 67 years), while the opposite pattern was observed in control participants. It thus appears that preserved or enhanced RSFC could play an important role in the brain health benefits associated with participation in musical activities in older age.

Our results specifically point towards a potential link between a history of musical activity and functional connectivity in the DMN, a critical network involved with normal and pathological neurocognitive aging and its risk states [28, 69, 70]. Interestingly, a recent training/intervention study reported an increased anticorrelation in RSFC between temporal brain regions (the right Heschl's gyrus) and the dorsal posterior cingulate cortex, as a key region of the DMN, after 12 months of piano training compared to control (musical listing awareness) intervention [71]. While we used a different study design, our results converge with the observation that musical activity is associated with higher RSFC partially involving DMN nodes. Albeit the temporal pole is not a core node of the DMN [27], it is considered part of a DMN subsystem that seems to be linked to higher-order cognitive processes including decision-making, memory, cognition, and problem-solving [72, 73]. On the other hand, the right precentral gyrus has been associated with motor processes [72, 74], while the right superior frontal gyrus has been reported to be involved in motor control tasks [73, 75]. Moreover, parts of this region appear to support working memory [76, 77] and to be anatomically connected with the DMN [78–80], placing its function between the temporal pole and the precentral gyrus. Increased functional connectivity between the MPFC, which is responsible for action and behavior regulation [81], and distant brain regions may represent a neural correlate of the complex sensory, motor and cognitive integration processes associated with playing a musical instrument.

Taken together, the present results are consistent with previous research in this field [21] and substantiate the notion that musical activities may help to preserve brain health in older age. The inherent multimodal stimulation associated with playing a musical instrument could enable the activation and integration of multiple brain regions and thus enhance neuroplasticity in musically active people [14–16]. This could promote a more efficient recruitment of neural resources and capacities, as suggested by previous studies in musically active younger and older adults [6, 82]. A recent musical training/intervention study investigating neural changes in musically non-experienced OA reported effects on functional connectivity (reflective of higher efficiency) between distant brain regions in a working memory task after 4 months of learning a musical instrument compared to passive control intervention [9]. Together, our and previous findings may imply that a history of musical activity might support the brain's ability to maintain and/or to facilitate functional connectivity or information transfer between specialized brain regions that are implicated in playing a musical instrument. Preserved or enhanced RSFC in these neural networks (both global and local RSFC) could support brain health and resilience in OA, as shown in previous studies investigating other reserve proxies [22, 31]. Prospective studies are needed to further clarify the neural correlates associated with long-term engagement in musical activity in OA and the extent to which they may be protective in normal and pathological neurocognitive aging.

In our study, lifetime musical activity was not significantly associated with local connectivity differences for the other higher-order brain networks, namely the SAL and FPN. This may be attributable to the triple network theory/model, which states that FPN and SAL in general show higher activation during tasks, while the DMN is most activated at rest [29]. Notably

though, a previous study has reported higher RSFC in nodes associated with the SAL network in younger musicians compared to non-musicians [83]. The difference between these findings and our observations could be explained by the different proficiency levels, recruitment/assessment methods, and age ranges of the respective study samples. Overall, a more detailed assessment of musical activity (for discussion see below) in population studies might be necessary to draw conclusions about the involvement of the FPN and SAL networks in the neural benefits associated with playing a musical instrument in older age.

### 4.3 Musical activity and global functional connectivity

Assessing global RSFC within and between the triple networks (DMN, SAL, and FPN) between OA with lifetime musical activity and matched controls, we found no evidence for significant differences between these two groups. Notably, a previous pilot training/intervention study in OA found in global RSFC within the FPN after a 4-month dance movement program compared to waiting list control [84], with dance being considered an enriching multimodal leisure activity with physical, cognitive, and social engagement [85]. It might thus be reasonable to assume that the cross-sectional nature of our study may not have sufficient power to detect presumably subtle differences in global RSFC patterns. In addition, more extensive information on the musical skills/experiences might be beneficial to detect a potential link between engagement in lifetime musical activity and global functional coupling within and between higher-order brain networks in OA.

### 4.4 Strengths and limitations

Overall, this study demonstrates that playing a musical instrument is associated with higher RSFC of the MPFC with distant frontal and temporal regions, which could be a neuroprotective correlate of this multimodal leisure activity in late life. The present results were found significant, adjusting for several other reserve proxies including years of education, crystallized intelligence, SES, and self-reported engagement in physical activity. Also, the two groups (with and without lifetime musical activity) did not differ in measures of lifetime educational and occupational enrichment and engagement in other leisure time activities. We incorporated a relatively large sample of well-characterized OA from the DELCODE cohort with high-quality neuroimaging data, collected using harmonized acquisition protocols across assessment sites. We further evaluated playing a musical instrument as an accessible leisure activity rather than a professional activity, as investigated in previous studies [68].

Following limitations need to be considered. 1. The present study used a cross-sectional design, which does not allow us to interpret the causality of the observed associations. Furthermore, the present sample size was modest. Our findings thus need to be replicated in future studies with larger samples to draw firm conclusions before longitudinal studies with rigorous musical training/intervention programs could provide deeper insights into the investigated relationships. 2. We used a one-to-one matching procedure to assign a matched control group to participants with lifetime musical activity. However, this procedure may introduce a potential arbitrary selection bias, which could affect the overall representativeness of the sample. In future or replication studies, a 1:1 randomization approach should rather be used for sample selection to better account for differences in confounding factors. 3. We considered a number of potentially confounding variables, including reserve proxies, which is a strength of our study. Nevertheless, other variables may be associated with a history of musical activity and might relate to variations in local and global functional connectivity patterns, including occupational, lifestyle, psychological and health-related characteristics. Albeit, we assessed several of these characteristics (such as self-reported social activity, personality traits, subclinical

depression and nutritional health status) and found no significant differences between the two groups, a comprehensive investigation of variables not included in the present RSFC analyses is warranted. 4. In the present study, musical activity was measured based on the self-reported frequency of playing a musical instrument over the lifetime, hence, the operationalization of this variable is not as ideal as it could be. More detailed information, e.g. on the type of musical instrument, playing solo or multiple instruments, age of acquisition, and/or participation in music groups, should be collected in cohort studies to better understand the association of these differences in musical instrument engagement on the observed neural benefits associated with musical activity in older age. This may also be a reason, why we were unable to detect associations between lifetime musical activity and global RSFC within the pre-selected RSN in the present study.

### 4.5 Conclusion

Our results indicate that playing a musical instrument during life relates to higher functional connectivity of the MPFC with frontal and temporal regions that are partially associated with the DMN. This finding suggests that preserved or enhanced RSFC between distant brain regions associated with higher-order cognitive and motor processes could be a functional neural correlate of the benefits associated with a history of musical activity. While preserved brain function may contribute to brain reserve and resilience in older age, further research is needed to investigate whether this might help delay cognitive decline and the onset of pathological conditions, including dementia. Overall, it can be concluded that the functional modulation/re-organization of higher-order brain networks could be a promising avenue to identify potential neuroprotective correlates underlying musical activity in OA.

## Supporting information

**S1 File.**
(DOCX)

## Acknowledgments

We are grateful to all DELCODE study teams at the various DZNE sites for their outstanding efforts. Furthermore, we would like to express our sincere gratitude to all volunteers and their family members, who participated in the DELCODE study. We are grateful for the invaluable contributions of the administrative and research staff in data collection, data management, and quality control. We thank Adriana Böttcher (Technische Universität Dresden, Germany) for her support in the design of this study. We are grateful for the methodological advice provided by Alexis Zarucha (DZNE, Dresden), Malo Gaubert (University of Rennes, France), and Dr. Daria Antonenko (University Medical Center Greifswald, Germany). We thank Prof. Shu-Chen Li (Technische Universität Dresden, Germany) for her insightful comments on data analysis and interpretation of results and Prof. Hans-Christian Jabusch (Hochschule für Musik Carl Maria von Weber Dresden, Dresden) for his valuable comments on the content of our manuscript. We further acknowledge the DELCODE study group: https://www.dzne.de/en/research/studies/clinical-studies/delcode/

## Author Contributions

**Conceptualization:** Maxie Liebscher, Katharina Buerger, Peter Dechent, Michael Ewers, Klaus Fliessbach, Wenzel Glanz, Stefan Hetzer, Daniel Janowitz, Ingo Kilimann, Christoph Laske, Falk Lüsebrink, Matthias Munk, Robert Perneczky, Oliver Peters, Josef Priller, Boris

Rauchmann, Ayda Rostamzadeh, Nina Roy-Kluth, Klaus Scheffler, Anja Schneider, Björn H. Schott, Annika Spottke, Eike Spruth, Jens Wiltfang, Frank Jessen, Emrah Düzel, Michael Wagner, Sandra Röske, Miranka Wirth.

**Data curation:** Lukas Preis, Stefan Teipel.

**Investigation:** Katharina Buerger, Peter Dechent, Michael Ewers, Klaus Fliessbach, Wenzel Glanz, Stefan Hetzer, Daniel Janowitz, Ingo Kilimann, Christoph Laske, Falk Lüsebrink, Matthias Munk, Robert Perneczky, Oliver Peters, Josef Priller, Boris Rauchmann, Ayda Rostamzadeh, Nina Roy-Kluth, Klaus Scheffler, Anja Schneider, Björn H. Schott, Annika Spottke, Eike Spruth, Jens Wiltfang, Frank Jessen, Emrah Düzel, Michael Wagner, Sandra Röske.

**Methodology:** Maxie Liebscher, Andrea Dell'Orco, Miranka Wirth.

**Project administration:** Lukas Preis, Stefan Teipel.

**Supervision:** Miranka Wirth.

**Writing – original draft:** Maxie Liebscher, Miranka Wirth.

**Writing – review & editing:** Maxie Liebscher, Johanna Doll-Lee, Miranka Wirth.

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
