## [Decision Letter · Decision Letter 0]

27 Nov 2023

PONE-D-23-20713Short Communication:

Lifetime Musical Activity and Resting-State Functional Connectivity in Cognitive NetworksPLOS ONE

Dear Dr. Liebscher, First, let me apologize on behalf of Plos ONE for the time needed to review this manuscript. It was very difficult to secure reviewers, also because a few reviewers with specific experience in the field were in conflict of interest. Reviewers 2 and 3 made a number of methodological comments that authors must consider, but not necessarily address by changing the processing strategy or study design. In particolar, I think that the "matched pair" experimental design is acceptable in this context; probably authors, rather than changing study design, can discuss the limitations of the approach in the relavant section of the manuscript.

Thank you for submitting your manuscript to PLOS ONE. After careful consideration, we feel that it has merit but does not fully meet PLOS ONE’s publication criteria as it currently stands. Therefore, we invite you to submit a revised version of the manuscript that addresses the points raised during the review process.

We look forward to receiving your revised manuscript.

Kind regards,

Federico Giove, PhD

Academic Editor

PLOS ONE

Journal Requirements:

4. Thank you for providing the following Funding Statement: 

“O. Peters received fees for consultation from Abbvie, Biogen, Eisai, Griffols, MSD Roche, and Schwabe. J. Priller received fees for consultation, lectures, and patents from Neurimmune, Axon, Desitin, and Epomedics. J. Wiltfang is an advisory board member of Abbott, Biogen, Boehringer Ingelheim, Immunogenetics, Lilly, MSD Sharp & Dohme, and Roche Pharma and received honoraria for lectures from Actelion, Amgen, Beeijing Yibai Science and Technology Ltd., Janssen Cilag, Med Update GmbH, Pfizer, Roche Pharma and holds the following patents: PCT/EP 2011 001724 and PCT/EP 2015 052945. J. Wiltfang is supported by an Ilidio Pinho professorship, iBiMED (UIDB/04501/2020) at the University of Aveiro, Portugal. E. Düzel received fees for consultation from Roche, Biogen, RoxHealth and holds shares in neotiv. F. Jessen received fees for consultation from Eli Lilly, Novartis, Roche, BioGene, MSD, Piramal, Janssen, and Lundbeck. The remaining authors report no conflicts of interest.”

We note that one or more of the authors is affiliated with the funding organization, indicating the funder may have had some role in the design, data collection, analysis or preparation of your manuscript for publication; in other words, the funder played an indirect role through the participation of the co-authors.

If the funding organization did not play a role in the study design, data collection and analysis, decision to publish, or preparation of the manuscript and only provided financial support in the form of authors' salaries and/or research materials, please review your statements relating to the author contributions, and ensure you have specifically and accurately indicated the role(s) that these authors had in your study in the Author Contributions section of the online submission form. Please make any necessary amendments directly within this section of the online submission form.  Please also update your Funding Statement to include the following statement: “The funder provided support in the form of salaries for authors [insert relevant initials], but did not have any additional role in the study design, data collection and analysis, decision to publish, or preparation of the manuscript. The specific roles of these authors are articulated in the ‘author contributions’ section.”

If the funding organization did have an additional role, please state and explain that role within your Funding Statement.

Please also provide an updated Competing Interests Statement declaring this commercial affiliation along with any other relevant declarations relating to employment, consultancy, patents, products in development, or marketed products, etc. 

5. PLOS requires an ORCID iD for the corresponding author in Editorial Manager on papers submitted after December 6th, 2016. Please ensure that you have an ORCID iD and that it is validated in Editorial Manager. To do this, go to ‘Update my Information’ (in the upper left-hand corner of the main menu), and click on the Fetch/Validate link next to the ORCID field. This will take you to the ORCID site and allow you to create a new iD or authenticate a pre-existing iD in Editorial Manager. Please see the following video for instructions on linking an ORCID iD to your Editorial Manager account: https://www.youtube.com/watch?v=_xcclfuvtxQ.

Reviewers' comments:

Reviewer's Responses to Questions

**Comments to the Author**

1. Is the manuscript technically sound, and do the data support the conclusions?

Reviewer #1: Yes

Reviewer #2: Partly

Reviewer #3: Partly

2. Has the statistical analysis been performed appropriately and rigorously? 

Reviewer #1: Yes

Reviewer #2: No

Reviewer #3: N/A

3. Have the authors made all data underlying the findings in their manuscript fully available?

Reviewer #1: Yes

Reviewer #2: Yes

Reviewer #3: Yes

4. Is the manuscript presented in an intelligible fashion and written in standard English?

Reviewer #1: Yes

Reviewer #2: Yes

Reviewer #3: Yes

5. Review Comments to the Author

Reviewer #1: RE: PONE-D-23-20713

The current manuscript describes an interesting and relevant cross-sectional study investigating the relationship between lifetime musical activity and resting-state functional connectivity (RSFC) in higher-order cognitive networks in the brain (i.e., the default mode network, frontoparietal, and salience networks).

Specifically, Dr. Liebscher and colleagues assessed n=130 cognitive unimpaired older adult individuals (age=>60 years) from the longitudinal DZNE-DELCODE cohort study on risk for dementia and cognitive decline. Individuals, who reported to have participated in musical instrument playing during early, middle, and late life stages (n=65) were compared to matched control subjects, who reported never to have played a musical instrument (n=65), in seed-to-voxel and within-networks RSFC patterns. Individuals with lifetime musical activity displayed higher local RSFC between medial prefrontal cortex, temporal, and frontal regions (right temporal pole and right precentral gyrus) versus control subjects.

The authors conclude that playing a musical instrument during life is related to a higher RSFC of medial prefrontal cortex with brain regions involved in higher-order cognitive and motor processes.

The manuscript is well-written and relevant. I only have the following minor points:

Abstract

- Since it is indeed a strength and relevant for interpretation of the observed findings that the two participant groups were matched on several demographic factors, please write these out already in the abstract (i.e., highlight that groups were matched on: age, sex, educational years, crystallized intelligence etc. (cf., Table 1))

Methods, 2.1 Participants:

- Please provide details on how you specifically operationalized “cognitively unimpaired”? It is unclear how this was assessed with standard neuropsychological tests. Providing information on the criteria for fulfilling this categorization (e.g., performing within certain cut-off values) would be beneficial.

Methods, 2.2 Measurements, 2.2.1 Self-reported assessment of musical activity:

- Although detailed information on the specific categorization process and the one-to-one matching procedure are provided in another REF and in the Supplementary Material, the manuscript would benefit from including these descriptions in the main text. Currently, it is unclear how often participants were to have played a musical instrument during life stages to be categorized as ‘having a lifetime history of musical activity’ (i.e., their musical activity ‘frequency’) from the only brief description provided in the current version of chapter 2.2.1. Please include these details from the reference and the supplementary materials here to enhance clarity. Indeed, this would equip the reader with enhanced ability to derive more nuanced conclusions of the identified association.

Methods, 2.2.4 Additional measures:

- How did you specifically operationalize socioeconomic status? Currently, it only says that SES was assessed using the International Socioeconomic Index of Occupational Information. Please specify what the specific items measure.

Discussion:

- Although the authors have ensured that there were no differences between musical activity participants and matched controls on several demographic variables (cf., Table 1), a more extensive examination of other unaccounted variables in the RSFC association analyses would be beneficial. Please provide a more comprehensive discussion of the potential factors that could also be linked to a history of musical activity and might have influenced the observed differences in local and global functional connectivity patterns (such as potential differences in functional capacity including autonomy, social activity/interpersonal relationships, and occupational functioning that may not be captured by the SES measure applied).

- Did the authors consider running sensitivity analyses with respect to individuals, who have reported only engaging in solo instrument performance, as compared to participation in musical bands as part of their lifetime musical activities? It would be of interest to explore this aspect and/or discuss the plausible associations potentially emanating from differences in musical instrument engagement on the identified neural benefits associated with participating in musical activities throughout life in older age.

- Additionally, it would be interesting to assess and/or discuss whether differences would emerge when accounting for the specific musical instruments played by the individuals (e.g., potential disparities between individuals engaged solely in vocal performance/singing, individuals exclusively playing instrument like the piano or violin, which additionally require intricate fine motor skills involving both the left and right hands). This would be particularly interesting with regards to the identified nodes of precentral gyrus and superior frontal gyrus found to be involved in motor processing and motor control tasks (cf., lines 362-365).

- Please also highlight that the samples sizes were modest and the findings need replication before firm conclusions can be drawn (before specifically highlighting the need for future longitudinal studies)

Figure 1

- In Figure 1C (FC cluster 1) and 1E (FC cluster 2), please provide value units on the Y-axis (currently, only values are reported)

Reviewer #2: I have some methodological concerns about this paper. Here are my comments:

1. The preprocessing of MRI is too concise. Please describe clearly what has been done. Since subjects were required to be registered to a template, the registration accuracy is very important for this seed-based analysis. Please demonstrate the quality of registration.

2. The seed-definition is too arbitrary. I think some studies have suggested the seed location of DMN is in around the PCC area. To confirm the accuracy to these seeds and determine if these seeds could be used to form networks, please generate network map based on these selected seeds.

3. Please give clear definition of local FC and global FC.

4. Statistical analysis is confusing. In the method section, three different threshold are mentioned as significant. For example, "p-values of < .05 were considered statistically significant", "using a voxel-

277 level threshold of p < .005 (uncorrected)", "a cluster extend threshold of p < .05 corrected for

278 multiple comparisons using False Discovery Rate". Please clarify the statistical significance level and ensure all the reported results could pass FDR.

5. Why not also compare the inter-network FC between two groups.

6. Image quality is not good.

Reviewer #3: The authors investigated the impact of playing musical instruments on the functional connectivity of cognitive neural networks. Utilizing the DELCODE Study database and employing a one-to-one matching procedure based on described characteristics, the authors compared FPN, DMN, SAL differences, revealing that "Higher RSFC has been related to protective lifestyle factors and preserved cognitive abilities in the context of brain pathology, supporting a contribution of these neural correlates to cognitive reserve". As suggested by the authors, the higher RSFC discovered in this study between higher-order cognitive and motor processes may represent neuroprotective correlates underlying musical activity in older adults (OA). However, the study design exhibits certain shortcomings. If possible, it is strongly recommended that the authors supplement essential data.

1. The issue of sample representativeness is raised. The text only mentions a one-to-one matching procedure, which introduces a considerable arbitrary selection component. It is strongly advised that the authors employ a 1:1 randomization approach to select samples, thus better eliminating differences in confounding factors (including those mentioned in the text, but not limited to only those) between the two groups.

2. It is strongly recommended that the authors supplement data on cognitive function, executive function, emotional changes, and relevant behavioral differences between the two groups in addition to the differences in resting-state brain functional connectivity. Calculating the correlation between brain functional connectivity differences and behavioral differences and conducting mediation analysis models or pathway analysis is preferable to establish persuasive evidence. It is crucial to demonstrate the specific aspects of cognitive function differences caused by the impact of playing musical instruments on the functional connectivity of cognitive neural networks. Therefore, the manuscript requires substantial data supplementation for a major revision.

3. The study lacks strict control over confounding factors. In addition to the factors mentioned in the text such as Age, Education, Crystallized Intelligence, Sex, Diagnostic group, Lifetime Physical Activity, Current Physical Activity, and LEQ, there may be other potential factors such as underlying diseases, nutritional status, emotional conditions, interpersonal relationships, etc. Adhering to the principle of random sampling is essential, and the sample size should ideally reach several thousand individuals.

6. PLOS authors have the option to publish the peer review history of their article (what does this mean?). If published, this will include your full peer review and any attached files.

Reviewer #1: No

Reviewer #2: No

Reviewer #3: No

---

## [Author Response · Author response to Decision Letter 0]

30 Jan 2024

Response to Reviewers

General comment to the reviewers and the academic editor

We thank the academic editor and reviewers for taking the time to read our manuscript and providing us with comments that have helped improve our work. With the assistance of the reviewers' feedback, we have revised our manuscript and incorporated current literature. The modifications are highlighted in yellow within the "Revised Manuscript with Changes Highlighted" document and are also referenced in our responses here.

Academic editor comments

General comment

“First, let me apologize on behalf of Plos ONE for the time needed to review this manuscript. It was very difficult to secure reviewers, also because a few reviewers with specific experience in the field were in conflict of interest.

Reviewers 2 and 3 made a number of methodological comments that authors must consider, but not necessarily address by changing the processing strategy or study design. In particolar, I think that the "matched pair" experimental design is acceptable in this context; probably authors, rather than changing study design, can discuss the limitations of the approach in the relavant section of the manuscript.

Thank you for submitting your manuscript to PLOS ONE. After careful consideration, we feel that it has merit but does not fully meet PLOS ONE’s publication criteria as it currently stands. Therefore, we invite you to submit a revised version of the manuscript that addresses the points raised during the review process.”

Response: We thank the academic editor for their time, patience and hard work in finding reviewers for our manuscript. We have addressed all of the reviewers' comments in this document and have revised our manuscript accordingly. We particularly appreciate your guidance regarding the critique of our matched-pair study design. We will also address all the points you raised below.

All comments

Comment #1

“1. Please ensure that your manuscript meets PLOS ONE's style requirements, including those for file naming. The PLOS ONE style templates can be found at

https://journals.plos.org/plosone/s/file?id=ba62/PLOSOne_formatting_sample_title_authors_affiliations.pdf“

Response: We thank you for bringing this to our attention and referring us to this helpful documentation. Based on these documents, we have changed the format and style of the headings, changed the naming of the figures in the text, modified the affiliation of the authors, and changed the names of the files. We have also ensured that we use one of the citation styles required by PLOS One (Vancouver).

Comment #2

“2. Did you know that depositing data in a repository is associated with up to a 25% citation advantage (https://doi.org/10.1371/journal.pone.0230416)? If you’ve not already done so, consider depositing your raw data in a repository to ensure your work is read, appreciated and cited by the largest possible audience. You’ll also earn an Accessible Data icon on your published paper if you deposit your data in any participating repository (https://plos.org/open-science/open-data/#accessible-data).”

Response: We thank you for sharing this interesting fact, but our data is particularly sensitive and personal data, which is why we cannot make it publicly accessible. However, it can be made accessible upon reasonable request. We have also uploaded our R scripts that were used for the data analyses to the open science framework.

Quote from online submission form and our manuscript on page 28: “Due to the sensitive and personal nature of the data, we are unable to publicly disclose the data. The anonymized data used for this study will be made available by request from any qualified investigator through the DZNE-DELCODE Steering Board for purposes of replicating procedures and results. Requests to access the minimal dataset should be directed to the German Center for Neurodegenerative Diseases (DZNE), Bonn. For contact information please refer to: https://www.dzne.de/en/research/studies/clinical-studies/delcode/ (Study Coordination and Project Management). We used existing data analysis packages for the neuroimaging and statistical analysis of this study. All relevant R scripts used for data analysis are freely available via the open science framework (OSF, https://osf.io/g73a9/).”

Comment #3

“3. We note that the grant information you provided in the ‘Funding Information’ and ‘Financial Disclosure’ sections do not match. When you resubmit, please ensure that you provide the correct grant numbers for the awards you received for your study in the ‘Funding Information’ section.”

Response: We appreciate you bringing this matter to our attention. Indeed, it seems there has been a misunderstanding, and we apologize for any confusion. The funding information referenced in your comment #4 does pertain to our competing interest. We would therefore be very grateful if the funding information in the online submission form could be replaced by the following statement:

Correct funding statement: “The overall DELCODE study was funded by the German Center for Neurodegenerative Diseases (Deutsches Zentrum für Neurodegenerative Erkrankungen [DZNE]), reference number: BN012. For the present study, the authors received no specific funding. The funders had no role in study design, data collection and analysis, decision to publish, or preparation of the manuscript.”

Comment #4

„4. Thank you for providing the following Funding Statement:

“O. Peters received fees for consultation from Abbvie, Biogen, Eisai, Griffols, MSD Roche, and Schwabe. J. Priller received fees for consultation, lectures, and patents from Neurimmune, Axon, Desitin, and Epomedics. J. Wiltfang is an advisory board member of Abbott, Biogen, Boehringer Ingelheim, Immunogenetics, Lilly, MSD Sharp & Dohme, and Roche Pharma and received honoraria for lectures from Actelion, Amgen, Beeijing Yibai Science and Technology Ltd., Janssen Cilag, Med Update GmbH, Pfizer, Roche Pharma and holds the following patents: PCT/EP 2011 001724 and PCT/EP 2015 052945. J. Wiltfang is supported by an Ilidio Pinho professorship, iBiMED (UIDB/04501/2020) at the University of Aveiro, Portugal. E. Düzel received fees for consultation from Roche, Biogen, RoxHealth and holds shares in neotiv. F. Jessen received fees for consultation from Eli Lilly, Novartis, Roche, BioGene, MSD, Piramal, Janssen, and Lundbeck. The remaining authors report no conflicts of interest.”

We note that one or more of the authors is affiliated with the funding organization, indicating the funder may have had some role in the design, data collection, analysis or preparation of your manuscript for publication; in other words, the funder played an indirect role through the participation of the co-authors. If the funding organization did not play a role in the study design, data collection and analysis, decision to publish, or preparation of the manuscript and only provided financial support in the form of authors' salaries and/or research materials, please review your statements relating to the author contributions, and ensure you have specifically and accurately indicated the role(s) that these authors had in your study in the Author Contributions section of the online submission form. Please make any necessary amendments directly within this section of the online submission form. Please also update your Funding Statement to include the following statement: “The funder provided support in the form of salaries for authors [insert relevant initials], but did not have any additional role in the study design, data collection and analysis, decision to publish, or preparation of the manuscript. The specific roles of these authors are articulated in the ‘author contributions’ section.” If the funding organization did have an additional role, please state and explain that role within your Funding Statement.

Please also provide an updated Competing Interests Statement declaring this commercial affiliation along with any other relevant declarations relating to employment, consultancy, patents, products in development, or marketed products, etc.

Within your Competing Interests Statement, please confirm that this commercial affiliation does not alter your adherence to all PLOS ONE policies on sharing data and materials by including the following statement: "This does not alter our adherence to PLOS ONE policies on sharing data and materials.” (as detailed online in our guide for authors http://journals.plos.org/plosone/s/competing-interests). If this

adherence statement is not accurate and there are restrictions on sharing of data and/or materials, please state these. Please note that we cannot proceed with consideration of your article until this information has been declared.”

Response: We are deeply appreciative of your comment. As previously explained in comment #3, the funding statement you referenced pertains to our competing interest. We apologize for any misunderstanding. The correct representation of our funding statement is as follows and we would be grateful if this could be corrected in the online submission form. In response to your comment, we have added a sentence regarding the role of the funder in this study. 

Correct funding statement: “The overall DELCODE study was funded by the German Center for Neurodegenerative Diseases (Deutsches Zentrum für Neurodegenerative Erkrankungen [DZNE]), reference number: BN012. For the present study, the authors received no specific funding. The funders had no role in study design, data collection and analysis, decision to publish, or preparation of the manuscript.”

We have also updated our competing interest as follows and would be grateful if this could also be updated on the online submission form:

Correct competing interest statement: “O. Peters received fees for consultation from Abbvie, Biogen, Eisai, Griffols, MSD Roche, and Schwabe. J. Priller received fees for consultation, lectures, and patents from Neurimmune, Axon, Desitin, and Epomedics. J. Wiltfang is an advisory board member of Abbott, Biogen, Boehringer Ingelheim, Immunogenetics, Lilly, MSD Sharp & Dohme, and Roche Pharma and received honoraria for lectures from Actelion, Amgen, Beeijing Yibai Science and Technology Ltd., Janssen Cilag, Med Update GmbH, Pfizer, Roche Pharma and holds the following patents: PCT/EP 2011 001724 and PCT/EP 2015 052945. J. Wiltfang is supported by an Ilidio Pinho professorship, iBiMED (UIDB/04501/2020) at the University of Aveiro, Portugal. E. Düzel received fees for consultation from Roche, Biogen, RoxHealth and holds shares in neotiv. F. Jessen received fees for consultation from Eli Lilly, Novartis, Roche, BioGene, MSD, Piramal, Janssen, and Lundbeck. The remaining authors report no conflicts of interest. This does not alter our adherence to PLOS ONE policies on sharing data and materials. “

Comment #5

“5. PLOS requires an ORCID iD for the corresponding author in Editorial Manager on papers submitted after December 6th, 2016. Please ensure that you have an ORCID iD and that it is validated in Editorial Manager. To do this, go to ‘Update my Information’ (in the upper left-hand corner of the main menu), and click on the Fetch/Validate link next to the ORCID field. This will take you to the ORCID site and allow you to create a new iD or authenticate a pre-existing iD in Editorial Manager. Please see the following video for instructions on linking an ORCID iD to your Editorial Manager account: https://www.youtube.com/watch?v=_xcclfuvtxQ.”

Response: We thank you for this information. We have added the ORCID iD to the PLOS ONE profile of the corresponding author.

Comment #6

“6. Your ethics statement should only appear in the Methods section of your manuscript. If your ethics statement is written in any section besides the Methods, please delete it from any other section.”

Response: We thank you for this comment and have included the ethics statement with the relevant information in our methods section and removed it from all other sections.

Quote from the manuscript, p. 6: “This study was based on the baseline dataset of the DZNE–DELCODE cohort, an ongoing German multicenter longitudinal study [32] designed and carried out in accordance with the ethical principles of the Declaration of Helsinki. The protocol was approved by local ethical committees at each study site and all participants provided written informed consent.”

Comment #7

“7. Please include captions for your Supporting Information files at the end of your manuscript, and update any in-text citations to match accordingly. Please see our Supporting Information guidelines for more information: http://journals.plos.org/plosone/s/supporting-information.”

Response: We thank you for this information. We have added the Supporting Information section to the manuscript as follows.

Quote from the manuscript, p. 21: “S1 File. Supplement: Lifetime musical activity and resting-state functional connectivity in cognitive networks. Contains supplementary methods and results as well as the following:

S1 Fig. Selection flow chart of our sample. The baseline DELCODE dataset included 1079 participants, and only participants with structural MRI data were included in the further selection process (n = 943). This study examined cognitively unimpaired participants; therefore, participants with mild cognitive impairment (MCI) and Alzheimer’s disease (AD) were excluded. In the next steps, only participants with complete data on lifetime musical activity assessment, matching variables, and covariates were included (total: n = 404). Finally, participants with available rs-fMRI and good-quality image data were considered for the matching process (total: n = 394). The final sample after the matching process consisted of n = 130 participants (participants with lifetime musical activity: n = 65, controls: n = 65). FH: family history of Alzheimer’s disease, MRI: magnetic resonance imaging, NC: normal cognition (healthy controls), rs-fMRI: resting-state functional magnetic resonance imaging, SCD: subjective cognitive decline, SES: socioeconomic status.

S2 Fig. Descriptive characterization of the included participants (n = 65) with lifetime musical activity across the three life stages. Information is based on participants’ responses about the frequency of musical activity in the Lifetime of Experiences Questionnaire (LEQ). The assessment of lifetime musical activity was - dependent on the participants’ age - assessed across two life stages (13-30 years and 30-65 years) or across three life stages (13-30 years, 30-65 years, 65 years and older).

S3 Fig. Result of the global (within-network) connectivity analysis. The ROI-to-ROI connectivity analysis within all three RSN showed no significant group differences. The brain maps (left side) show the DMN (A.), the FPN (C.) and the SAL (E.) networks with all investigated ROIs. Note, ROIs displayed in dark grey specify the network seeds used in our local (seed-to-voxel) connectivity analysis. The corresponding graphs (right side) show the non-significant difference between older participants with lifetime musical activity and matched controls in their global resting-state connectivity (z scores) within the DMN (B.), the FPN (D.) and the SAL (F.). Box plots display the median with 95% confidence intervals, interquartile range with lower (25th) and upper percentiles (75th), and individual data points within each group. Key: ACC: Anterior Cingulate Cortex, AInsula (L): Anterior Insula left, AInsula (R): Anterior Insula right, DMN: Default Mode Network, FC: Functional connectivity, FPN: Fronto-Parietal Network, LP (L): Lateral parietal left, LP (R): Lateral parietal right, LPFC (L): Lateral Prefrontal Cortex left, LPFC (R): Lateral Prefrontal Cortex right, MPFC: Medial Prefrontal Cortex, ROI: Regions-of-interest, PCC: Posterior Cingulate Cortex, PPC (L): Posterior Parietal Cortex left, PPC (R): Posterior Parietal Cortex right, RSN: Resting-State Networks, SAL: Salience Network.

S1 Table. Network-specific ROIs with their MNI coordinates for the calculation of the global RSFC (for graphical representation see S3 Fig.). MNI: Montreal Neurological Institute; ROIs: regions-of-interest; RSFC: resting-state functional 

---

## [Decision Letter · Decision Letter 1]

20 Feb 2024

Short communication

Lifetime musical activity and resting-state functional connectivity in cognitive networks

PONE-D-23-20713R1

Dear Dr. Liebscher,

We’re pleased to inform you that your manuscript has been judged scientifically suitable for publication and will be formally accepted for publication once it meets all outstanding technical requirements.

Kind regards,

Federico Giove, PhD

Academic Editor

PLOS ONE

Additional Editor Comments (optional):

Reviewers' comments:

Reviewer's Responses to Questions

**Comments to the Author**

1. If the authors have adequately addressed your comments raised in a previous round of review and you feel that this manuscript is now acceptable for publication, you may indicate that here to bypass the “Comments to the Author” section, enter your conflict of interest statement in the “Confidential to Editor” section, and submit your "Accept" recommendation.

Reviewer #1: All comments have been addressed

Reviewer #2: All comments have been addressed

2. Is the manuscript technically sound, and do the data support the conclusions?

Reviewer #1: Yes

Reviewer #2: Yes

3. Has the statistical analysis been performed appropriately and rigorously? 

Reviewer #1: Yes

Reviewer #2: Yes

4. Have the authors made all data underlying the findings in their manuscript fully available?

Reviewer #1: Yes

Reviewer #2: Yes

5. Is the manuscript presented in an intelligible fashion and written in standard English?

Reviewer #1: Yes

Reviewer #2: Yes

6. Review Comments to the Author

Reviewer #1: The authors have satisfactorily responded to the comments and suggestions raised by the undersigned, which has improved the quality of the manuscript. I therefore recommend that the manuscript should be accepted for publication and wish to congratulate the authors on this piece of work.

Reviewer #2: (No Response)

7. PLOS authors have the option to publish the peer review history of their article (what does this mean?). If published, this will include your full peer review and any attached files.

Reviewer #1: No

Reviewer #2: No
